# Deep Machine Learning for Path Length Characterization Using Acoustic Diffraction

**Brittney Erin Jarreau** [1,*,†] **and Sanichiro Yoshida** [2,†]

1   U.S. Naval Research Laboratory, Stennis Space Center, Bay St. Louis, MS 39529, USA
2   Department of Chemistry and Physics, Southeastern Louisiana University, Hammond, LA 70402, USA
*   Correspondence: brittney.jarreau@nrlssc.navy.mil
†   These authors contributed equally to this work.

**Abstract:** Many fields now perform non-destructive testing using acoustic signals for the detection of objects or features of interest. This detection requires the decision of an experienced technician, which varies from technician to technician. This evaluation becomes even more challenging as the object decreases in size. In this paper, we assess the use of both traditional signal-processing machine learning algorithms, Long Short-Term Memory (LSTM), as well as Convolutional Neural Network (CNN) architectures to approximate acoustic anomalies with an eye toward micro-scale applications such as application to biofilms. The probing signal is generated using a continuous sound wave emitted at controlled frequencies of 1 and 5 MHz through metallic specimens of varying heights each containing an anomaly in the form of a hole. Data are collected as the transmitted signal is sampled at several locations as the wave travels through the specimen. We have developed both a CNN and an LSTM architecture for frequency-domain feature detection and approximation. The CNN models, one for phase and one for amplitude data, take short-distance Fourier transforms (SDFTs) representing the change in the signal over multiple observation points as input. The LSTM model takes the change in phase or amplitude points at each lateral location as a comma-separated value (CSV) input. The models analyze the frequency and spatial changes experienced by each specimen and produce an estimation of the acoustic path length of the anomaly in radians. The models are evaluated using mean-square error and the R-square statistic. All models perform with a fairly high R-square score, the amplitude CNN and LSTM models achieving upwards of a 99% fit and the phase CNN achieving a 97% fit on average for the predicted values. With the performance of these models, we demonstrate that utilizing the transfer function phase and amplitude data to analyze acoustic diffraction patterns leads to the ability to extract, with great precision, features in the input signal that describe the nature of the anomaly.

**Keywords:** acoustics; convolutional neural network; long short-term memory; machine learning; deep learning

## 1. Introduction

An acoustic wave is the flow of pressure and velocity patterns as sound propagates through a medium. Such waves are often observed to be longitudinal, meaning the medium moves in the direction of the wave. However, under various circumstances, an acoustic wave traveling through a solid medium can exhibit transverse modes instead. With this change, the medium begins to move in a direction perpendicular to the wave. This induced movement of the medium results in a wave that carries information that can be used to gain insight into the structure of the medium. Specifically, this acoustic energy is telling of the elastic property of the medium and can be used to probe the structure of the specimen [1].

Acoustic waves have long been utilized across several fields to probe for details of the structure of materials [2,3]. Examination of acoustic data is often accomplished by reviewing the reflection of the acoustic signal as a two-dimensional image. Such techniques typically

employ ultrasonic waves with frequencies ranging from 500 kHz to 10 MHz. In many cases, ultrasonic testing techniques have proven to be more efficient than rival non-destructive testing methods such as visual inspection, thermography, and shearography at detecting Barely Visible Impact Damages (BVIDs) [4]. One such method is acoustic microscopy [5]. Acoustic microscopy is a form of ultrasonic testing used to detect microscopic anomalies. Thanks to such advancements, hidden and microscopic anomalies are able to be imaged for non-destructive testing.

Sound waves are observed to exhibit longer wavelengths than those seen in light waves. Because of this, achieving high spatial resolution in acoustic data is difficult. Typical acoustic analysis is limited to analysis of the amplitude of the received acoustic signal. This is problematic in cases of low acoustic reflectivity, where the acoustic impedance of the anomaly and the surrounding environment are too similar. In such cases, there will be a low contrast in the resulting image.

One case of interest in which low acoustic contrast can be anticipated is the acoustic probing of biofilms. Biofilms are communities of microorganisms that accumulate via a substance called extracellular polymeric substance (EPS) at an interface between two mediums [6,7]. Imaging biofilms is not a trivial task as the cells which contribute to the biofilm have a very similar acoustic impedance to the surrounding fluid, as both the bacterial cells and surrounding fluid are composed mainly of water [8]. The compounding of low resolution and contrast yields poor-quality images, which are very difficult for humans to interpret.

Acoustic signals are characterized by two main components: amplitude and phase. Amplitude is used to understand the magnitude of energy carried by an acoustic wave. Phase, on the other hand, provides the acoustic path length. When both the original wave and the received wave are known, the two sets of data can be used together to get a more clear picture of the system. This is accomplished by calculating the transfer function as the change in the emitted signal exhibited by the received signal. Doing so illuminates the change that an acoustic signal undergoes when passing through a system. Previous research on using the phase, amplitude, transfer function phase, and transfer function amplitude to detect an acoustic anomaly shows that the transfer function phase and the transfer function amplitude provide more useful information on the nature of the anomaly than phase and amplitude alone [9]. The improved quality of information is due to the transfer function filtering out components that are common to both the emitted and received signal, thus allowing examination of just the changes due to interaction with the medium being probed.

Biofilms are so thin that the resultant diffraction falls in the Fresnel range where the lateral profile of the amplitude and phase varies in a complicated fashion. This makes manual analysis of the diffraction patterns a difficult task as human eyes cannot make clear determinations. In this research, we aim to determine whether a machine can extract the necessary features to determine the depth. Depth is an important feature because it tells us the elasticity of the anomaly as the acoustic path length is the product of the elastic modulus and the physical length. In this study we assess the feasibility of assessing acoustic diffraction data by creating a model which aims to make determinations in both the Fresnel range and Fraunhofer range. To combat the previously introduced challenges of analyzing acoustic imagery, the transfer function phase and transfer function amplitude of the signal are analyzed. This solves the issue of low reflectivity commonly experienced in acoustic imagery. Using this data, the machine learning algorithms compensate for low spatial resolution so that common image processing techniques can be applied to the acoustic data. This enables the machine learning algorithm to locate the anomaly and extract more detailed information which characterizes the acoustic path length of the anomaly.

The preliminary experiment is performed on thin metallic specimens with induced anomalies of several dimensions, as described in Section 2. Designing the model in such a way allows more control over the diffractive properties and acoustic path lengths of the ground truth data that the model will train on, thus allowing a controlled analysis of the model's performance with known data to be performed. After the desired performance is

achieved by this preliminary model, transfer learning can then be used to adapt the model for use with biofilm specimens. The results from this preliminary experiment lend weight to using this technique to distinguish characteristics of biological materials such as biofilms. Two areas in which this would be particularly useful are in determining the presence and thickness of a biofilm and in determining the density of a biofilm. One complication is that the elastic constant of a cell and that of water are similar, making it difficult to distinguish constituent cells from the surrounding fluid. This results in low acoustic contrast and reflectivity [10] in the imagery. This preliminary research, however, has shown that a CNN will be able to make determinations even in low-contrast acoustic imagery, so this should not be problematic for a machine learning approach.

The body of this article is organized into the following sections: introduction, preprocessing, materials and methods, results, discussion, and conclusions. The current section details the problems with biofilm detection, which motivate the current study as well as the theory which drives the model development. The next section, Section 2, demonstrates the setup of the experiment, the data collection process, the data preprocessing, and the model design. Section 3 details the data analysis used to assess the applicability of each model, the performance of the models during training, and the performance of the models when used to predict acoustic path length on an additional dataset. Section 4 examines the performance of each model as it relates to the theoretical concepts introduced in the current section as well as the analysis performed in Section 3. The final section, Section 5, wraps the discussion up with conclusions that can be drawn from Section 4 and recommendations for future work to address some of the shortcomings of the model discussed in that section.

### 1.1. Motivation

This research draws motivation from the necessity to produce a methodology for assessing the health of biofilms in situ. The formation and spread of biofilms has been a long-standing challenge in the medical field as well as industrial settings. These biofilms often lead to the contamination of delicate systems and corrosion of industrial systems [6]. The EPS in which biofilms form allows microbes to form large colonies where they undergo physiological changes that increase their resistance to chemical challenges [11]. This often leads to the need for multiple treatment routes, particularly in the medical field. The health of the biofilm, and therefore susceptibility to treatment, can be understood based on the density and uniformity of the biofilm [11]. Most methods of studying biofilms require destructive methodologies or staining of the microbes [11,12]. Newer techniques use microscopic approaches to study the biofilms; however, these approaches require the biofilm to be spread uniformly on flat surfaces, which is not conducive to an in situ study [11,12].

In 2010 a study was performed to understand how the growth of biofilm impacts acoustic properties in porous media [13]. The authors collected acoustic samples over 29 days and found that this time, the signals began the same for the individual and stimulated samples, but near day 29, the signals changed. The lower-frequency components begin to arrive later while the high-frequency components begin to arrive earlier [13]. This research indicates that the acoustic properties of the biofilm are clearly exhibited in the resulting acoustic signal. In 2015 a study was conducted on the early detection of biofilm growth in immunocompromised patients, which demonstrated the increased complexity of analyzing biofilms in situ [14]. The authors demonstrated that the acoustic impedance of the biofilm (1.9 MRayl) is so similar to that of human soft tissue (1.35 to 1.85 MRayl), that imaging the biofilm using ultrasound in-vivo was too difficult using ultrasound alone. The authors saw a large improvement after coupling the high-frequency acoustic microscopy with targeted ultrasound contrast agents to enhance the images to overcome the low acoustic contrast [14]. As the average soft-tissue density is roughly the same as water, the same complications are likely to arise when trying to determine the health of a biofilm [15].

While the similar acoustic impedance of biofilm components to their surroundings may lead to low acoustic contrast, one property that may be exploited to account for the difference in signal strength and arrival time seen by Davis et. al. [13] is the acoustic

path length of the constituent cells in the biofilm. The change in acoustic velocity after Day 29 indicates that the acoustic velocity becomes dependent on the frequency. This means the media between the source and the receiver begins to demonstrate properties of dispersion. Since the water is unchanged, we can interpret all changes in the signal as the column becoming dispersive [13]. Normal dispersion is the dispersion where the wave velocity decreases with the frequency, so the dispersion exhibited by the column is normal dispersion. The acoustic path length is the product of the physical distance from the source to the receiver and the wave number (the reciprocal of the wavelength). This physical distance is unchanged regardless of the time. On the other hand, being proportional to the frequency, the acoustic path length shows the above effect of dispersion. Thus, it would be reasonable to use a property such as acoustic path length for detecting constituent biofilm cells.

Similar to the work in [13], we aim to develop an approach for analyzing biofilms using properties of acoustics. Our research differs, however, in that we aim to conduct this analysis by using the property of acoustic diffraction. Biofilm density is typically approximately 1.14 g/cm$^3$, with a thickness ranging from 50 to 100 µm, depending on the species [12]. At such a thickness and perceivable size, Fraunhofer diffraction is unlikely. This means that any acoustic analysis for biofilm studies is likely to be interpreted in the more complex, Fresnel range. It is necessary then, to understand how an acoustic diffraction model can be expected to perform on estimating the acoustic path length of small anomalies in thin specimens to determine how well analysis of acoustic diffraction data for characterization of acoustic anomalies can be expected to perform.

### 1.2. Theory

To perform the necessary assessment of how machine learning algorithms perform when producing an estimate of acoustic path length, we use a surrogate experiment with which acoustic diffraction data can be more quickly and cost-efficiently produced and analyzed. In this experiment we compare two machine learning approaches for using acoustic diffraction data collected on thin metallic specimens in both the Fresnel and Fraunhofer regimes to estimate the acoustic path length of the anomaly. Producing the ability to estimate such a complex feature is an important step toward enabling the automated estimation of density and thickness of biofilm specimens. The experiment is carried out using steel rings as shown in Figure 1 and the dimensions of each ring used can be found in Table 1. The controlled anomalies for which the acoustic path length will be calculated are holes with diameters of 6.47 mm and 2.17 mm.

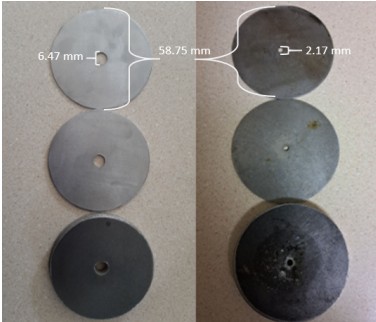

**Figure 1.** The two sets of rings used in this experiment with the three 6.47 mm anomalies on the left and the three 2.17 mm anomalies on the right. Each set of rings contains 3 different thicknesses: 0.68 mm (top), 1.32 mm (middle), and 6.46 mm (bottom).

**Table 1.** Dimensions of specimens used in data collection.

| Disc Height (mm) | Disc Diameter (mm) | Anomaly Width (mm) |
| --- | --- | --- |
| 0.68 | 58.75 | 6.47 |
| 0.68 | 58.75 | 2.17 |
| 1.32 | 58.75 | 6.47 |
| 1.32 | 58.75 | 2.17 |
| 6.46 | 58.75 | 6.47 |
| 6.46 | 58.75 | 2.17 |

The steel rings are placed in the center of the experiment, directly over the transmitter on the bottom. The transmitter emits signals at 1 and 5 MHz in order to increase the number of acoustic path lengths induced in the experiment. Data are then collected from a steel observation plate on top of the specimen where the receiver sweeps from left to right to pick up the transmitted signal. This setup can be seen in Figure 2. A layer of water is kept between all components to increase the acoustic coupling of the system. More details on the experimental setup can be found in Section 2.

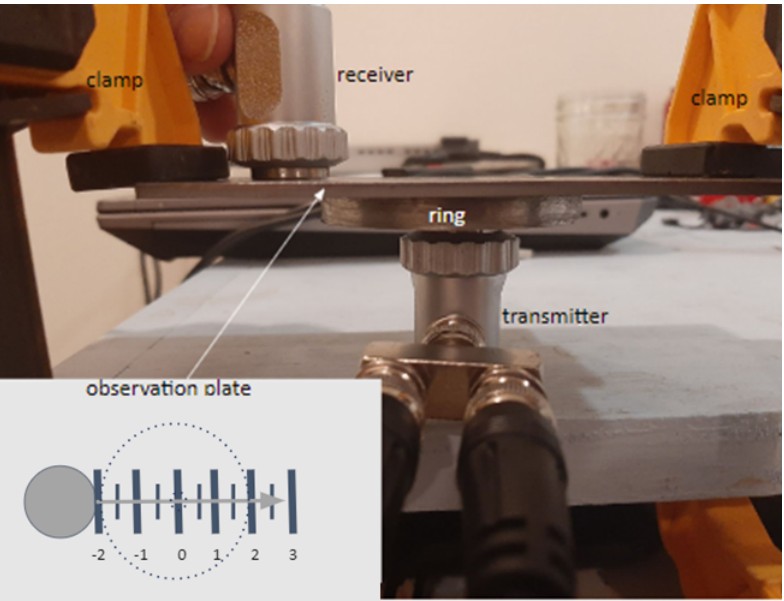

**Figure 2.** The experimental setup used for data collection. In the main portion of the figure we see the emitter at the bottom of the system with the steel ring directly on top followed by a steel observation plate and the receiver on top of that. In the depiction of the observation plate in the lower left corner, the receiver (solid circle) is shown at its starting position of −2 cm and the arrow across the center depicts the motion of the sweep. A trace of the steel specimen that lies beneath the plate is depicted with the dotted circle to give perspective on the position of the sweeping receiver relative to the specimen. The transmitter, shown as the bottom component in the main image, lies directly beneath the center of the steel specimen.

Figure 3 depicts the acoustic wave traveling from an acoustic emitter into a medium containing a known acoustic anomaly and then to the receiver on top. In this figure, the axis X denotes the direction in which the receiver scans across the observation plate. The axis Z denotes the axis of propagation for the acoustic signal. The receiver detects the signal as it scans the observation plane. At the location of the anomaly, the acoustic wave is partially reflected. Due to this reflection, the wavefront is deformed by the time it is picked up by the receiver. The change in the wavefront depends on the shape of the

interface and the acoustic impedance of the anomaly. In many situations, these are generally unknown; however, in this experiment, the acoustic impedance of the anomaly and size of the anomaly are controlled. Because of this, we can approximate the change experienced due to acoustic impedance. The surface of the emitter is PZT and the medium used in this experiment is steel. Within the steel is an anomaly in the form of a hole drilled through the center. As a layer of water is kept between all components, including the transducer and the steel ring and between the steel ring and steel observation plate on top of that ring, it is common for the water to end up in the hole. Because of these factors, the acoustic impedance at the anomaly is determined by the properties of PZT, air, steel, and water at room temperature since these are the media through which the acoustic wave passes. More on the experimental setup can be found in Section 2. The acoustic impedance of each media is calculated using the following formula [16]. The results of this calculation can be seen in Table 2.

$$Z = density * velocity \tag{1}$$

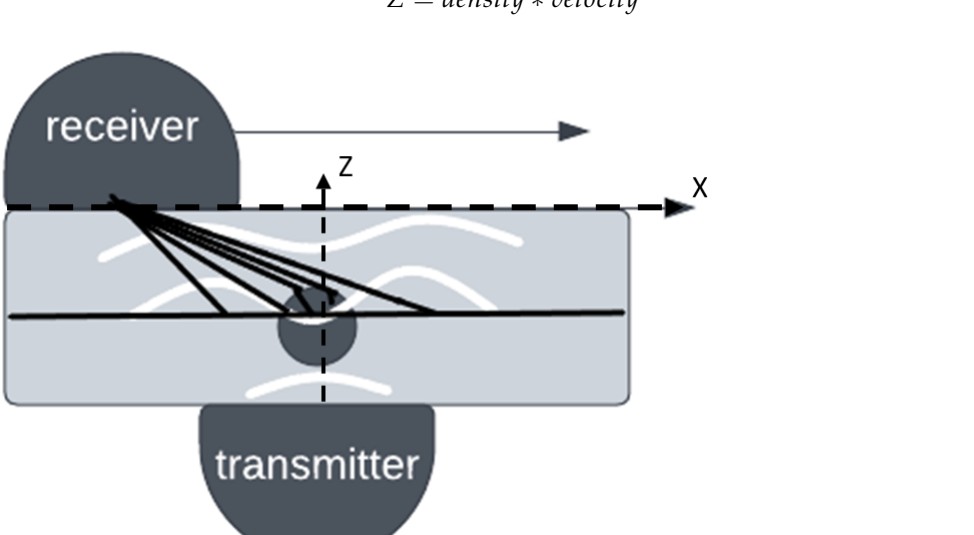

**Figure 3.** Depiction of an acoustic wave traveling through anomaly in a solid material. The white lines of the figure depict the wave prior to interaction with the anomaly and after diffraction by anomaly. The black lines depict the signal captured by the sensor, represented as element waves. The horizontal dotted line labeled X denotes the axis along which the receiver sweeps/scans the specimen. The vertical dotted line labeled Z denotes the axis of propagation of the acoustic signal.

Using the values from Table 2 and the formula for the reflectance of a wave at the interface between two media as the wave travels in a nearly straight path, we can approximate the expected reflection to be as follows. On entrance into the anomaly leaving the transducer, we examine both the scenario where the wave travels from PZT into water and where the wave travels from PZT into air:

$$PZT - Water = \frac{(Z_w - Z_p)}{(Z_w + Z_p)} = \frac{(31.5 \times 10^6 - 1.481 \times 10^6)}{(31.5 \times 10^6 + 1.481 \times 10^6)} \approx -0.91 \tag{2}$$

$$PZT - Air = \frac{(Z_a - Z_p)^2}{(Z_a + Z_p)^2} = \frac{(31.5 \times 10^6 - 398.89)}{(31.5 \times 10^6 + 398.89)} \approx -1.00. \tag{3}$$

**Table 2.** Values for derivation and acoustic impedance of media involved in this experiment.

| Medium | Density (kg/m$^3$) | Velocity (m/s) | Impedance (Rayls) |
|--------|--------------------|----------------|---------------------|
| Water | 1000 | 1481 | $1.481 \times 10^6$ |
| Air | 1.204 | 331 | 398.89 |
| Steel | 7850 | 5940 | $46.63 \times 10^6$ |
| PZT | 7500 | 4200 | $31.5 \times 10^6$ |

We can interpret this calculation as meaning that at least 91% of the wave as it enters the anomaly will reflect back toward the emitter. For the incident portion of the wave which enters into the hole, we then consider the acoustic impedance of this wave as it exits the hole toward the steel observation plate:

$$Steel - Water = \frac{(Z_s - Z_w)}{(Z_s + Z_w)} = \frac{(46.63 \times 10^6 - 1.481 \times 10^6)}{(46.63 \times 10^6 + 1.481 \times 10^6)} \approx 0.938 \tag{4}$$

$$Steel - Air = \frac{(Z_s - Z_a)^2}{(Z_s + Z_a)^2} = \frac{(46.63 \times 10^6 - 398.89)}{(46.63 \times 10^6 + 398.89)} \approx 1.00. \tag{5}$$

From the above calculations, we are able to infer that upwards of 93.8% of the wave which was transmitted through the hole will be reflected at the anomaly as the wave will either travel between steel and water or steel and air. This means that nearly all of the signal will reflect at the location of the hole, so nearly all of the wave will be bending around the obstacle at the center. This will lead to evidence of an acoustic shadow behind the hole. The nature of this acoustic shadow is determined by the size of the obstacle, the wavelength of the signal, and the distance to the observation point [17].

As mentioned in the introduction, data for this experiment are captured with two frequencies, 5 MHz and 1 MHz. The steel specimens being probed also contain 2 possible anomaly sizes, 6.47 mm in the larger anomaly cases and 2.17 mm in the smaller anomaly cases. These specimens are always observed at a distance of 6.47 mm, on the other side of the steel observation plate. Diagrams estimating the effect of these configurations on the observation of the acoustic shadow are depicted in Figure 4. These diagrams depict the anticipated acoustic shadow which will occur directly over the obstacle depicted in Figure 3 as the obstacle blocks a portion of the received signal. As we see in the figures, theoretically, the acoustic shadow should be apparent in both of the 1 MHz cases as the point of observation falls well within the acoustic shadow depicted by the gray triangle, but may be more difficult to observe or even missed in the 5 MHz cases where the observation point falls nearly at the tip of the anticipated shadow. This is hinted at by the observation distance and width of the shadow shown by the dotted line in Figure 4.

In addition to the acoustic shadow, the divergence of the wave as it passes around the obstacle varies depending on the largest dimension of the obstacle and the observation point. When the distance between the observation point and the obstacle is much greater than the largest dimension of the obstacle, the diffraction is Fraunhofer diffraction [18,19]. The further into the Fraunhofer range the observation point moves, the more Fraunhofer-like the observed wavefront becomes. When the distance to the observation point is comparable or less than the largest dimension of the obstacle, the diffraction is Fresnel diffraction [20]. As with the Fraunhofer range, the further into the Fresnel range the observation point moves, the more Fresnel-like the observed wavefront becomes. Similarly, as the observation point approaches the boundary between Fresnel and Fraunhofer diffraction, the wavefront is observed to be less Fresnel-like or less Fraunhofer-like. The problem with approximations in the Fresnel domain is that this form of diffraction is not a simple function. In the Fresnel range the shape and intensity of the diffraction pattern take many forms as the wave propagates downstream due to the increased scattering, while in the Fraunhofer

range the shape and intensity remain rather constant [20]. This is shown in Figure 5 as the Fresnel regime, the region prior to the Fraunhofer diffraction limit (R), begins with a rectangular pattern and evolves into a pattern with many peaks as it approaches R. The calculations of the Fraunhofer limit (R) are displayed in Table 3, which are calculated using the following equation.

$$R = D^2 * \lambda, \text{where D is the largest dimension of the anomaly} \qquad (6)$$

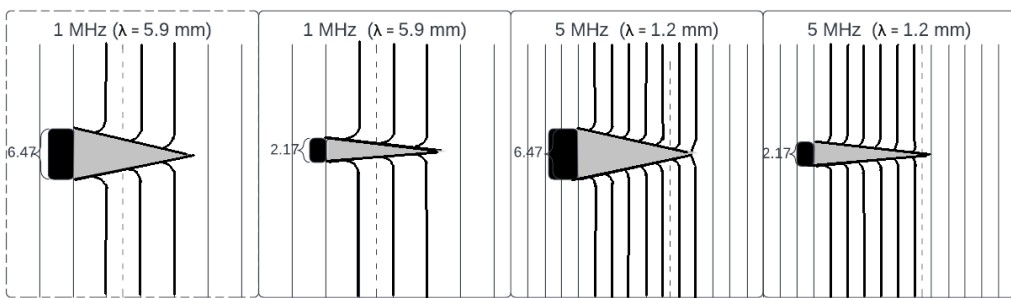

**Figure 4.** This figure depicts the observation point (depicted as a dotted line) in relation to the anticipated acoustic shadow the gray triangle for the four configurations used in this experiment. This figure is a theoretical representation of the expected acoustic shadow as the wave begins to bend around an obstacle, represented by the black box, as more wavefronts, represented by the vertical partitions, pass the obstacle. Here we note the width of the acoustic shadow at the point of observation, which is decreasing from the left-most figure to the right-most figure.

**Table 3.** Values for derivation of and Fraunhofer diffraction limit.

| Frequency | Anomaly Width (mm) | Anomaly Height (mm) | Wavelength (mm) | Diffraction Limit (mm) |
|---|---|---|---|---|
| 1 MHz | 6.47 | 0.68 | 5.9 | 7.09 |
| 1 MHz | 2.17 | 0.68 | 5.9 | 0.79 |
| 1 MHz | 6.47 | 1.32 | 5.9 | 7.09 |
| 1 MHz | 2.17 | 1.32 | 5.9 | 0.79 |
| 1 MHz | 6.47 | 6.46 | 5.9 | 7.09 |
| 1 MHz | 2.17 | 6.46 | 5.9 | 7.07 |
| 5 MHz | 6.47 | 0.68 | 1.2 | 34.88 |
| 5 MHz | 2.17 | 0.68 | 1.2 | 3.92 |
| 5 MHz | 6.47 | 1.32 | 1.2 | 34.88 |
| 5 MHz | 2.17 | 1.32 | 1.2 | 3.92 |
| 5 MHz | 6.47 | 6.46 | 1.2 | 34.88 |
| 5 MHz | 2.17 | 6.46 | 1.2 | 34.77 |

The Fresnel diffraction pattern takes on many forms between the zones shown in this figure, only two Fresnel zones are shown here to reduce the complexity of this image [21]. On the other hand, the Fraunhofer regime maintains a very similar shape as it begins at R and moves further away. In this research, we observe anomalies in both the Fresnel and Fraunhofer regime, as can be seen by the change in proximity of the red dotted line in Figure 5. This figure depicts the changes in the diffraction pattern of an acoustic signal as the wave propagates away from the origin.

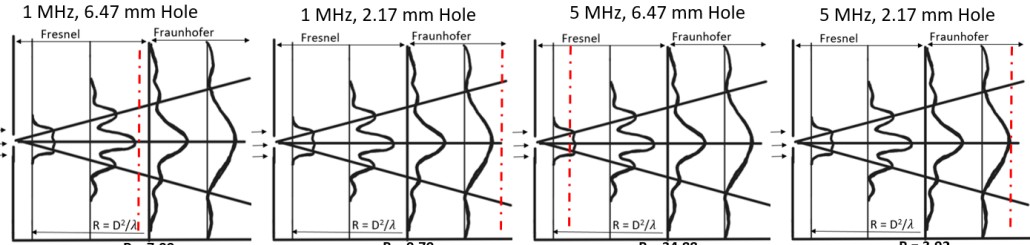

**Figure 5.** The limit between Fresnel and Fraunhofer diffraction for several experiments is given as R. The relationship of the observation point of 6.47 mm (the dotted line) can be examined in relation to R (the central line) to estimate the level of complexity of the diffraction pattern which will be observed in the data.

Due to the complexities of multiple forms of diffraction and multiple diffraction ranges, it is difficult to produce an analytical function for the diffraction profile of the acoustic signal. Although an analytical expression is not easily expressed, the signal at a given observation point may still be characterized. According to the Huygens-Fresnel theory [22], every point on a wavefront is a source of spherical wavelets which mutually interfere. As such, the acoustic signal received at the top of the experiment can be expressed as the superposition of all secondary waves that initiated at the entrance plane of the acoustic anomaly. This concept is illustrated by the black lines of Figure 3.

The Huygens-Fresnel theory allows us to presume that the sensor signal observed at any lateral location can be represented as the amplitude and phase of the acoustic wave at that location. The observed amplitude and phase for each point are those of the superposed waves at those points [23]. Since these values are directly impacted by the shape, size, and acoustic impedance of the anomaly, the collected signal contains information about the nature of the anomaly. This tells us that the spatial profile of the observed wave is uniquely determined by the properties of the anomaly which caused the diffraction.

As discussed in [9], additional complexity is introduced as the signal crosses the barrier between the emitter's surface and the steel specimen. This is due to a portion of the signal being reflected back toward the emitter. The reflection must be removed to improve the quality of the received signal. This is not easily accomplished in the time domain; however, in the frequency domain, we can eliminate this effect by using the wave's transfer function. To calculate this transfer function, the FFT of the received wave is divided by the FFT of the original wave, this too is presented in [9]. The transfer function amplitude represents the reduction in the acoustic energy as it passes through the system while the transfer function phase indicates the change in acoustic path length. Together the amplitude and phase describe the full structural change of the wave caused by the medium. By analyzing the transfer function, we can extract features of an anomaly while ignoring the reflection barriers that are constant in the system. Therefore, the proposed machine learning methodologies utilize the transfer function's phase and amplitude data for analyzing the anomalies.

Theoretically, it seems we can characterize the anomaly using the x-dependence of the sensor signal. In actuality, however, these signals are not easily differentiated. Figure 6 exhibits spatial profiles of the transfer function phase and amplitude data in which the anomaly size and acoustic path length varies. The human eye struggles to differentiate the different cases based on the spatial profile of this sensor data, particularly for the phase data which appear nearly random. It is easiest to see the similarities for matching acoustic path lengths among the amplitude data while very few similarities are found for the phase data. Finding trends in the data becomes more complex as the number of configurations increases. This hints at the need for machine learning to look for more complex features.

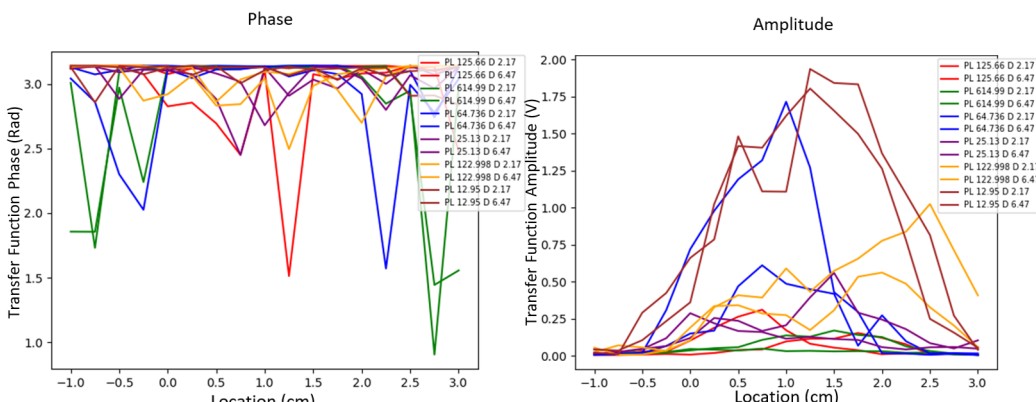

**Figure 6.** The profile of the transfer function phase data (**left**) and transfer function amplitude data (**right**). One sample for each combination of anomaly size (D) and acoustic path length (PL) is presented. In each chart, the profiles corresponding to the same acoustic path length are plotted in the same color to help analyze for similarities which correlate to acoustic path length.

## 2. Materials and Methods

The data to be used by the models are collected using a Hantek 6000BE USB oscilloscope with a Koolerton 15 MHz DDS Signal generator. The signal was emitted in two forms: as a controlled sine wave with a frequency of 5 MHz and sampled at a rate of 100 MHz, and as a controlled sine wave with a frequency of 1 MHz and sampled at a rate of 25 MHz. The transducers used for the 1 MHz data are Yushi 1 MHz, 24 mm ultrasonic Straight Beam Probe NDT transducers (S16144), which have a beam waist (beam radius at the transducer surface) of 12 mm. The 5 MHz transducers are Yushi 5 MHz, 14 mm ultrasonic Straight Beam Probe flaw detector transducers (S60216), which have a beam waist of 7 mm. The metallic specimen to be examined sits at the center of the system. In this experiment, the specimen is a steel ring. Several variations of the steel ring are examined ranging in thickness and anomaly size, these can be seen in Figure 1. The acoustic wave used to probe the specimen is emitted from directly below with the specimen's hole over the center of the transmitter. A layer of water is kept between transducers as well as the specimen and observation plate to ensure acoustic coupling. The probing signal is sampled at 17 locations on the observation plate. The signals are collected in a sweeping fashion, collecting several signals at each point along the length of the specimen. This section describes the experimental design, the data collection process, the data preparation, and the architecture of each model.

### 2.1. System Design and Data Collection

Data are generated by emitting continuous sound waves with controlled frequencies of 1 and 5 MHz through metallic specimens of varying heights each containing anomalies. The Hantek oscilloscope is used in combination with two sets of Yushi ultrasonic transducers. The Yushi 1 MHz ultrasonic Straight Beam Probe NDT transducers (S73224) have a beam waist of 12 mm, while the Yushi 5 MHz ultrasonic Straight Beam Probe flaw detector transducers (75,815) have a beam waist of 7 mm. The 1 MHz transducers have a beam waist of 12 mm and the 5 MHz transducers have a beam waist of 7 mm. Adding this variation in wavelength and beam waist allows an increased variability in the dataset which will lead to better generalizations of the data. The anomalies induced are holes with diameters of 6.47 mm and 2.17 mm; the rings can be viewed in Figure 1 and the dimensions of the rings and anomalies are specified in Table 1. In this experiment, the bottom transducer, the transmitter, is connected between the generator and channel 1 of the oscilloscope to capture the wave being sent into the medium. The top transducer, the receiver, is used to collect the wave on the opposite end of the medium. The experimental setup used to gather data for the models is shown in Figure 2.

The goal of the experiment is to determine if machine learning methodologies can find significant differences in the diffraction pattern caused by an acoustic anomaly which provide information on the acoustic path length of the anomaly. To control the sampling points, a steel observation plate is used. This plate has markings at each half-centimeter from the center of the plate to the edges of the specimen, covering roughly 6 cm in total. Several measurements are taken at each point, aligning the right edge of the receiver to each marking on the plate. Measurements are also taken halfway between each point, aligning the receiver's right edge at the center of two points. Measurements are always taken sliding from the left-most marking to the last marking on the right side of the plate, which provides data ordered from −1 to 3 cm. Each set of movements from left to right accounts for a single trial. For each specimen, four trials are collected. The oscilloscope software stores each waveform, the transmitted and the received waveform, to the host computer as a CSV file containing the sampling rate and the wave voltage. Each pair of CSV files are then consolidated into a single CSV which contains a column for time steps, initial voltage, and received voltage.

The transmitted signal is sampled at 17 lateral locations, each 0.25 cm apart. For 4 rounds of data collection, 5 signals are captured at each of the 17 locations. The 5 signals per location vary slightly in position at the location of interest to capture the variation in data collection, which would naturally occur between different operators. These 5 signals are combined at random to create 625 different lateral profiles for examination by the model. The data collected could have been combined to create $5^{17}$ sets of waveforms; however, that is far more data than are needed to train this model. Training on such a large amount of data would likely lead to a lack of generalization as the patterns in the data are very similar. To avoid this, we limited the dataset generation to 625 combinations of the collected signals. Additionally, these profiles are saved in reverse, which simulates a slight shift of the specimen from left to right. In this way, 5000 signals per specimen are collected. Since data are collected on 6 specimens per hole diameter and at 2 frequencies, this leads to a database of 60,000 sets of waveforms per acoustic path length (regression target) to be processed. This database building process is summarized in the pseudocode of Figure 7.

```
for frequency in [1, 5]:              <-- 2 * 30000 = 60000 files
  for diameter in [6.47, 2.17]:       <-- 2 * 15000 = 30000 files
    for height in [0.68, 1.2, 2.17]:  <-- 3 * 5000 = 15000 files
      for trial in [1...4]:           <-- 4*1250 = 5000 files
        while i < 625:                <-- 2*625 = 1250 files
          for location in [1...17]:
            add_to_array(signals_at_location[get_random(1...5)])
          save_array_to_file(array)
          save_reverse_array_to_file(array)
```

**Figure 7.** Pseudocode representing the process of taking all 5 collected signals at each of the 17 locations in all 4 rounds of data collection on each of the 6 specimens at both frequencies and combining them to create 60,000 sets of waveforms in the form of an array to be processed into images.

*2.2. Preprocessing*

Data preprocessing begins by randomly selecting 17 waveforms from a given round of data collection. The fast Fourier transform (FFT) is computed on the emitted and received signal for each of the 17 waveforms. While in the Fourier domain, the transfer function amplitude and transfer function phase are calculated as these values give insight into the changes that the wave has undergone as it travels through the medium. The transfer functions are calculated by taking the Fourier Transform of the received signal and dividing it by the Fourier Transform of the input signal. This allows us to remove a degree of noise from the phase and amplitude features in our training. Noise as the signal leaves the transmitter often occurs due to factors such as poor contact between the emitter and the specimen and fluctuations in the buffer water between the transducer and the ring.

If only the received signal is analyzed, these factors which are not useful in probing the specimen are included in the signal. However, if the transfer function is taken, such factors can be removed so that a cleaner signal is analyzed. The transfer function phase and amplitude data for all frequency ranges are then grouped into 51 bins with the maximum value representing the bin so that all neighboring frequencies are combined. As described in [9], once all FFTs are taken and binned, they are pieced together as columns of an array, such that the array has 51 rows by 17 columns. At this point, the remainder of the signal preprocessing differs depending on which model the data will go to.

In addition to preprocessing the signals, a portion of the control data must also be preprocessed to convert the disc parameters from height to acoustic path length. The equation used to calculate the expected acoustic path length is shown below and the derivations of acoustic path length for each combination of specimen height and frequency can be found in Table 4.

$$\text{Path Length} = 2\pi \frac{\text{disc height}}{\lambda} \tag{7}$$

**Table 4.** Values for derivation of the 6 acoustic path lengths.

| Frequency | Wavelength (mm) | Height (mm) | Path Length (Rad) |
|:---------:|:---------------:|:-----------:|:-----------------:|
| 1 MHz | 0.33 | 0.68 | 12.95 |
| 1 MHz | 0.33 | 1.32 | 25.13 |
| 1 MHz | 0.33 | 6.46 | 123.0 |
| 5 MHz | 0.066 | 0.68 | 64.74 |
| 5 MHz | 0.066 | 1.32 | 125.66 |
| 5 MHz | 0.066 | 6.46 | 614.99 |

2.2.1. Convolutional Neural Network

As convolutional neural networks are designed to process images, the preprocessing of the data for this type of model begins with converting the received waveforms into an image. In our previous work on using machine learning for acoustic diffraction patterns, we introduced the concept of Short Distance Fourier Transforms (SDFTs) [9]. These SDFTs represent the change in the waveforms over both frequency and distance in the form of an image so that the data can be parsed by a CNN. Each SDFT, in this case, is created by taking the fast Fourier transform (FFT) of each of the 17 waveforms collected at each lateral location for a single trial. For example, the first trial has data collected in 2.5 mm increments at locations from −1 to 3 cm. Once all FFTs are taken, an FFT for each location is selected randomly from the set of 5 waveforms collected for each location in each Trial. Each of these FFTs are then stacked into an image as a column of the SDFT image in order from −1 to 3 cm. These images are saved as greyscale images and stored in directories that correspond to the data type (amplitude or phase), hole diameter, driving frequency, and thickness of the disc. To enable easy use with a machine learning algorithm, the file names and important metadata, such as the experiment, data type, hole diameter, trial number, acoustic path length, and disc thickness are stored in a CSV file so that any value can be pulled out to use as a training label for the images. Examples of the SDFTs are included in Figure 8.

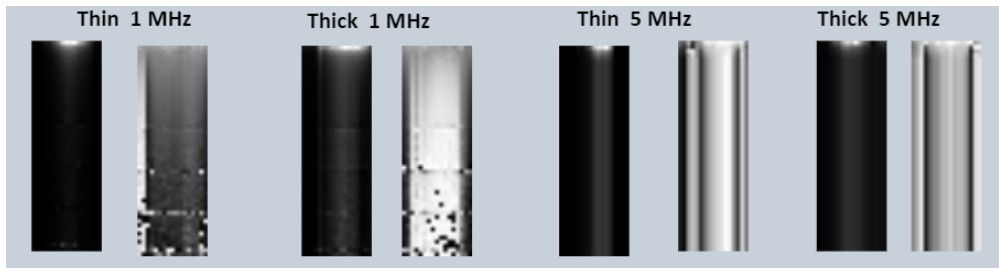

**Figure 8.** Pairs of SDFTs are shown for several samples from the database representing changes in the transfer function amplitude (**left**) and transfer function phase (**right**).

### 2.2.2. Long Short-Term Memory Neural Network

LSTM models expect to receive samples in sequential order. As we are mainly interested in examining the transfer function phase and amplitude data, the LSTM model uses only the bins which correspond to the transfer function phase and amplitude at the driving frequency of 1 or 5 MHz. To put this information in a format applicable to a LSTM model, the lateral positions must be represented as a sequence of samples rather than a collection of samples corresponding to differing positions. To accomplish this, the Phase and Amplitude value for each lateral location is added as an independent row in a CSV along with the experiment the data correspond to, the hole diameter, the trial number, the acoustic path length, and the thickness of the disc. Many of these features will not be used by the LSTM but are useful to include so that the dataset can be used in other models in the future. In this way, a CSV is created such that each group of 17 consecutive rows corresponds to a single amplitude and phase profile. A sample of the Acoustic Path Length, Amplitude, and Phase arrangement in the CSV is shown in Figure 9.

| | Acoustic Path Length | Phase | Amplitude |
|---|---|---|---|
| 0 | 125.660 | 3.131213 | 0.002489 |
| 1 | 125.660 | 3.134429 | 0.002983 |
| 2 | 125.660 | 3.117818 | 0.001195 |
| 3 | 125.660 | 3.109684 | 0.005953 |
| 4 | 125.660 | 2.669541 | 0.006915 |
| ... | ... | ... | ... |

**Figure 9.** Reduced CSV of LSTM data shows how phase and amplitude are arranged for LSTM to easily consider each lateral step as a subsequent sample.

For this experiment, LSTM and CNN architectures were explored. Both models examine the spatial relationship of changes in the transfer function phase and amplitude data and the relationship of those changes to acoustic path length. For the CNN architecture, the phase and amplitude data were separated into two separate models to examine if phase or amplitude yields a better approximation of acoustic path length. For the LSTM model, phase and amplitude are kept together for each lateral observation point. This section details the structure of each model.

### 2.3. Convolutional Neural Networks

#### 2.3.1. Background

The Convolutional Neural Network (CNN) architecture builds on its predecessor, Artificial Neural Network (ANN), to maintain spatial relationships which are typically seen in image data. As with ANNs, CNNs are made up of layers of neurons that use non-linear functions to optimize their weights, thus learning relationships within the data [24]. Unlike ANNs, CNNs are able to take in images in a three-dimensional representation, which allows the network to maintain the relationship between rows and columns of data. Because of this property of CNNs, they are typically used in image processing and pattern recognition

problems, although they may be applicable to any problem which has a clear relationship between two dimensions of data [24,25]. For instance, in 2020 Sharma and Kumar applied a CNN to non-image-based breast cancer data. They did this by transforming the data into bar graphs, distance matrices, and normalized numeric values and feeding these images or 2-D arrays into the CNN as the input data. The authors achieved 94–100% accuracy on the breast cancer dataset, proving even non-image-based data can be successfully analyzed using CNNs [25].

The general components of a CNN model are convolution layers, pooling layers, and fully connected layers [26]. The two-dimensional data are passed to the CNN's input layer where it goes through these components and finally outputs the predicted values using features extracted along the way.

The first layer through which the data travel is the convolution layer. This layer determines the output through the activation of regions of the input data using the scalar product between the weights of the neurons and the region [24]. The weight vector is commonly referred to as a filter or kernel. It is applied in a sliding fashion so that the scalar product is calculated between the weight vector and all input components [26]. The typical activation function used at this layer is the rectified linear units (ReLU) function. This one is commonly selected as it corrects negative values to a value of 0, which leads to the removal of these weights. This is evident from the ReLU formula in Equation (8). This operation extracts a specified number of features from the input data [26].

$$ReLU(x) = \begin{cases} 0 & \text{if } x < 0 \\ x & \text{if } x >= 0 \end{cases} \tag{8}$$

After leaving a convolution layer, the extracted features are typically sent to a pooling layer. At this point in the model, the location of the extracted features is no longer significant [26]. As such, down-sampling can be performed to reduce the number of features the model must keep up with [24]. Additionally, performing max-pooling helps to increase generalizations in the learned features by providing an abstract representation of the features. Few pooling techniques exist, but the most common are average pooling and max pooling. These differ in the way the resulting value is calculated to represent the contributing data components. Max-pooling represents the data in the pooling window with the maximum of all values while average pooling uses the average of the contributing values [26].

Finally, after some number of convolution and pooling layers, the resulting matrices are flattened into vectors and passed to fully connected layers. These layers perform the same duties as ANNs to make decisions about the data [24]. In the fully connected layers, the dot product of the weight vector and input vector, flattened features, is computed. This operation can be optimized using various batch sizes and optimizers [26]. One powerful optimizer is the Adaptive Moment Estimation (ADAM) optimizer [27]. ADAM is a computational and memory-efficient algorithm that does well at preserving the learning gradient. ADAM combines the AdaGrad's ability to deal with sparse gradients [28] and RMSProp's ability to deal with non-stationary objectives [29]. It has been proven to be well-suited for big-data problems [27].

### 2.3.2. Amplitude CNN

The CNNs created for this data each have a similar ten-layer design. The amplitude model trains on the SDFTs representing the change in amplitude. As seen in the preprocessing analysis, the amplitude relationship does not look as complex as the phase relationship; therefore, this model has a decreased complexity in comparison with the phase model. The model design includes an input layer, four convolutional layers, two max-pooling layers, a flattening layer, two fully connected layers, and an output layer. All input data (pixels and acoustic path lengths) are normalized between zero and one. This section details the layers of this model.

**Input**: The CNN takes in 51 × 17 × 1 images in the form of SDFTs. This means images which are 51 pixels tall, 17 pixels wide, and in grayscale. The values in the SDFTs are normalized between 0 and 1. Examples of the SDFTs are shown in Figure 8.

**Convolution Layer 1 & 2**: The first two convolution layers apply 64 kernels which are sized 7× 7 and use a stride of 1 with the same size padding. The same size padding will pad 0's around the exterior of the image so that the kernel can be applied to all pixels of the image without reducing the size of the image. This is important in this model as the images are very narrow, but the relationships are fairly complex.

**Max Pooling 1**: The first two convolution layers are followed by a max pooling layer with a kernel-sized 2 × 2. The use of such a max pooling layer allows us to reduce the dimensionality of the model while maintaining the most important features from each kernel. Max pooling also increases the generalizations made by the model to reduce noise in the image.

**Convolution Layer 3**: This convolution layer applies 64 5 × 5 kernels with a stride of 1 and same size padding.

**Convolution Layer 4**: This convolution layer applies 96 kernels which are sized 5 × 5 and use a stride of 1 with the same size padding.

**Max Pooling 2**: The second two convolution layers are followed by a max pooling layer with a kernel sized 2 × 2 to once again reduce the dimensionality and allow generalization of the extracted features.

**Flattening**: Flattening layers take the resulting feature maps from the final max pooling layer and flatten them into a feature vector to allow connection to a dense layer. Immediately after flattening, 40% of the features are dropped to avoid overfitting the features in the feature vector.

**Fully Connected Layers**: The flattening layer connects to a fully connected, or dense, layer with 400 nodes.

**Output**: The fully connected features are connected to a single node which uses a regression with the hyperbolic tangent function to determine the acoustic path length of the input sample. This function works best for the Amplitude model as the relationship between the transfer function amplitude and the normalized acoustic path length is a nonlinear relationship which is increasing between 0 and 1.

The architecture for the amplitude model is depicted in Figure 10.

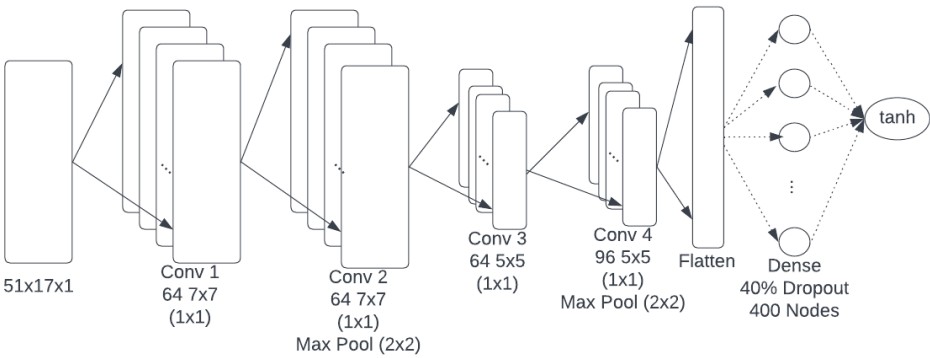

**Figure 10.** The Model Architecture for the Amplitude CNN.

2.3.3. Phase Model

The phase CNN uses a similar ten-layer design to the amplitude model. The model trains on the SDFTs representing the change in phase. Preprocessing analysis showed low and varying correlations between the lateral locations hinting at a more complex phase

relationship. The model design includes an input layer, four convolutional layers, two max-pooling layers, a flattening layer, two fully connected layers, and an output layer. All input data (pixels and acoustic path lengths) are normalized between zero and one. This section details the layers of this model.

**Input**: As with the Amplitude model, this CNN takes in $51 \times 17 \times 1$ images in the form of SDFTs. In this case, the SDFTs represent transfer function phase data. Again, the images are 51 pixels tall, 17 pixels wide, and in grayscale and normalized between 0 and 1.

**Convolution Layer 1 & 2**: The first two convolution layers apply 64 kernels which are sized $7 \times 7$ and use a stride of 1 with the same size padding. The same size padding will pad 0's around the exterior of the image so that the kernel can be applied to all pixels of the image without reducing the size of the image. This is important in this model as the images are very narrow, but the relationships are fairly complex.

**Max Pooling 1**: The first two convolution layers are followed by a max pooling layer with a kernel-sized $2 \times 2$. The use of such a max pooling layer allows us to reduce the dimensionality of the model while maintaining the most important features from each kernel. Max pooling also increases the generalizations made by the model to reduce noise in the image.

**Convolution Layer 3**: This convolution layer applies 96 $5 \times 5$ kernels with a stride of 1 and the same size padding.

**Convolution Layer 4**: This convolution layer applies 96 kernels which are sized $5 \times 5$ and use a stride of 1 with the same size padding.

**Max Pooling 2**: The second two convolution layers are followed by a max pooling layer with a kernel sized $2 \times 2$ to once again reduce the dimensionality and allow generalization of the extracted features.

**Flattening**: Flattening layers take the resulting feature maps from the final max pooling layer and flatten them into a feature vector to allow connection to a dense layer. Immediately after flattening, 40% of the features are dropped to avoid overfitting the features in the feature vector.

**Fully Connected Layers**: The flattening layer connects to a fully connected layer with only 230 nodes.

**Output**: The fully connected features are connected to a single node which uses a regression with the ReLU function to determine the acoustic path length. This function works best for the Phase model as the relationship between the transfer function amplitude and the normalized acoustic path length is a more linear relationship which is increasing between 0 and 1.

The architecture for the phase model is depicted in Figure 11.

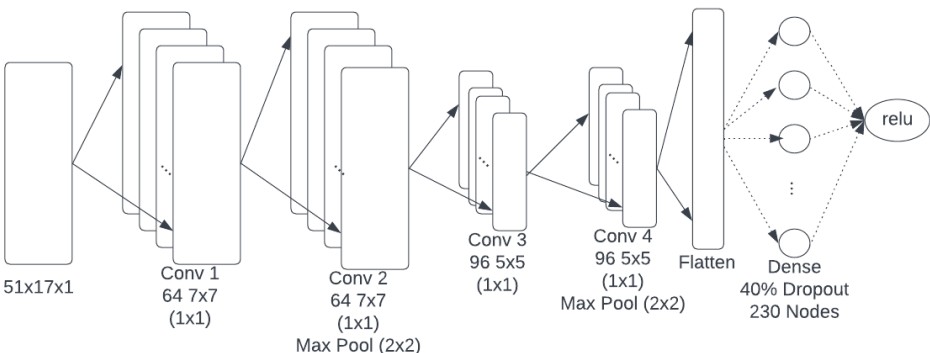

**Figure 11.** The Model Architecture for Phase CNN.

Both models use ReLU as the activation function in the convolution layers. ReLU is the standard choice of activation function for convolution layers, as this activation function sets all negative neurons to 0. This has the effect of removing black pixels and replacing them with gray pixels, which helps to reduce contrast in the features. Each model is also optimized using an Adam optimizer with a learning rate of 0.0001. The Adam optimizer was selected as it can rapidly optimize performance by quickly seeking out the local minima while tuning the weights of the model. The learning rate of 0.0001 with these models gave the best trade-off of training time and model performance. The models also use mean square error as the loss function, which is fairly standard in regression models. Finally, the models train using a batch size of 128 for 20 epochs or until the loss begins to stabilize.

## 2.4. Long Short-Term Memory Neural Network

The Long Short-Term Memory (LSTM) architecture, similar to the CNN architecture, builds on ANN to maintain relationships across multiple samples. The first step in this direction was the development of Recurrent Neural Networks (RNNs). RNNs are neural networks that utilize context units to allow the network to use time-delayed feedback for maintaining relationships between samples [30]. RNNs introduce the ability to maintain contextual information about the mapping between input and output data. However, RNNs have a limited scope of context they can utilize due to a vanishing gradient. LSTM improves the architecture by replacing the context units of the RNN with memory blocks. Each memory block contains interconnected memory units as well as an input, output, and forget gate. The use of these gates enables the LSTM memory cells to store and access information over long periods of time maintaining relationships with high importance and removing relationships with low importance. This solves the vanishing gradient problems seen in RNNs for large data [31].

### Phase-Amplitude Model

The LSTM used in this experiment has five layers. This design includes an input layer, three LSTM layers, and an output layer. All input data (amplitude, phase, and acoustic path lengths) are normalized between zero and one. The layers of the model are detailed in this section.

**Input**: The LSTM accepts input as seventeen pairs of phase and amplitude values in the form of a vector sized $17 \times 2$. These have been normalized between 0 and 1 and represent the change in phase and amplitude over all lateral locations for each sample collected.

**LSTM Layers 1 and 2**: The first two LSTM layers have a memory size of 17 units allowing the model to remember 17-time steps. Each of these layers also uses a hyperbolic tangent activation function.

**LSTM Layer 3**: The second LSTM layer connects to a third layer with only 9 memory units so that only the most important half of the LSTM's memory is considered at the last layer. This layer again uses a hyperbolic tangent activation function.

**Output Layer**: For the output layer, the final LSTM layer connects to an output layer with a single node. This layer is a dense layer that performs a regression to the ReLU activation function. The ReLU activation function led to the best performance when compared with the hyperbolic tangent function for the phase and amplitude pairs.

The model is optimized using an Adam optimizer with a learning rate of 0.001. The learning rate of 0.001 with these models gave the best trade-off of training time and model performance. The model also uses mean square error as the loss function. The model is depicted in Figure 12.

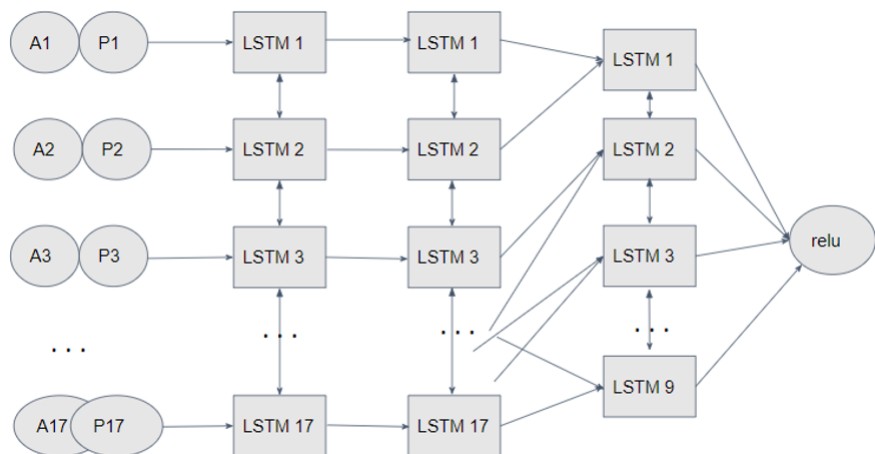

**Figure 12.** The Model Architecture for LSTM.

### 3. Results

In verifying that the data collected are applicable to both CNN and LSTM architectures, it is important to ensure that the distribution of the data corresponds to possible relationships between each sample. For this purpose, we examine the box-and-whisker charts in Figure 13. We see that in the amplitude data the values have the most variation near the central locations; however, the range of outliers remains fairly consistent across all locations. This means that outliers likely correspond to mild changes in the experimental setup due to variations in the positioning of the receiver, rather than anomalies in the data, which would make patterns in the data difficult to learn. The higher variation near the center is likely due to the increased strength of the signal near the center of the plate. On the converse, we see that there is more variation in phase values as the data are taken further from the center. This is because phase is evaluated as the ratio of the sine component to the cosine, which decreases as we move toward the edge of the system. The smaller the values, the greater the error. In the phase data, the range of the outliers mostly decreases as we move toward the center of the system except for a few locations within one centimeter of the center. This is likely due to the rapid change in phase, which may be observed in various locations near the center depending on where the anomaly is located. This will lead to more complexities in learning the trend of the transfer function phase data.

Finally, we examine the correlation of the phase and amplitude data across all lateral locations and with the acoustic path length values. In Figure 14, we can see there is a strong correlation between each amplitude value with its neighboring value, however, only mild correlations between neighboring phase values. This is expected because the amplitude is a smoother function of x than the phase. We also see a much stronger correlation of the middle amplitude values with acoustic path length while phase only exhibits strong correlations with acoustic path length at the exterior locations. Despite the lower correlations in the phase locations, it appears there is enough correlation that the spatial relationships between values will be learnable by an LSTM.

For all models, performance was assessed by observing the improvement of loss as mean square error over all training epochs, by plotting the actual vs. the predicted acoustic path lengths, and by producing the R-square metric. Examining each of these metrics enables us to examine how the models performed, comparatively.

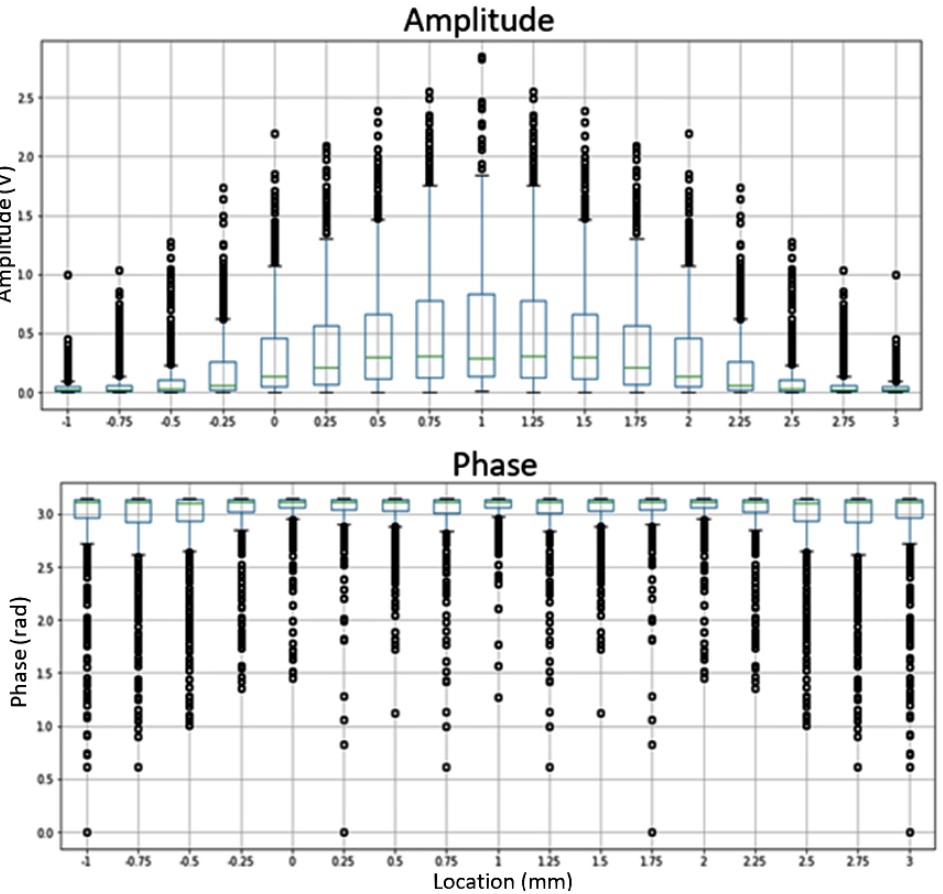

**Figure 13.** Examining the data range and distribution of amplitude (**top**) and phase (**bottom**) values.

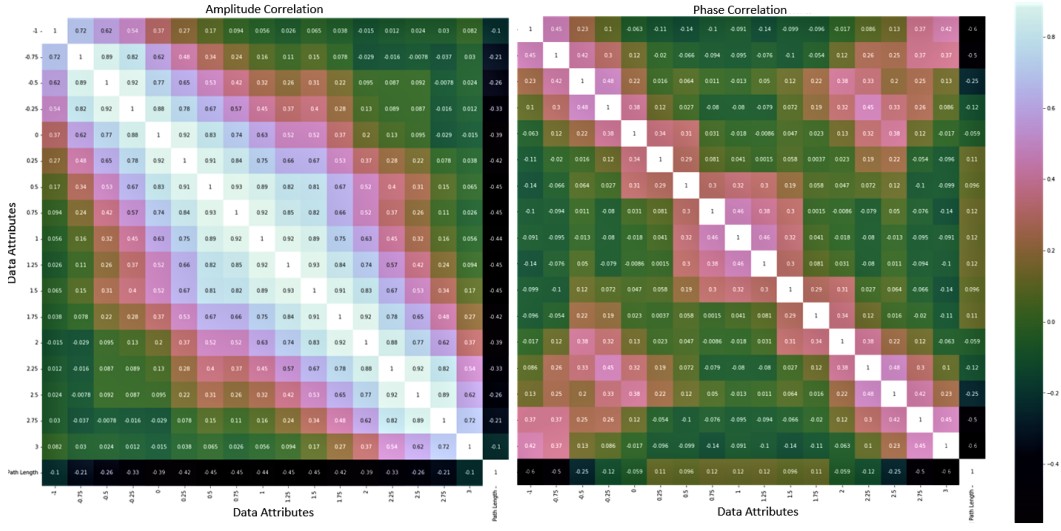

**Figure 14.** Correlation matrices of Amplitude values (**left**) and phase values (**right**).

From the trend in the mean square error (MSE), we can determine that the model has trained to a stable point and has minimized the mean square error efficiently and without over-fitting. In Figure 15, it is evident that the training and validation error begins roughly the same for each model. From here, the LSTM struggles to decrease at first, before rapidly descending around epoch 9 and converging near 0.00067. The phase CNN model's error decreases smoothly as the model trains for 8 epochs, after which the loss begins to stabilize near 0.003. Finally, the amplitude CNN experiences a large drop in loss immediately and spends the rest of the epochs slowly converging near an MSE of 0.00016.

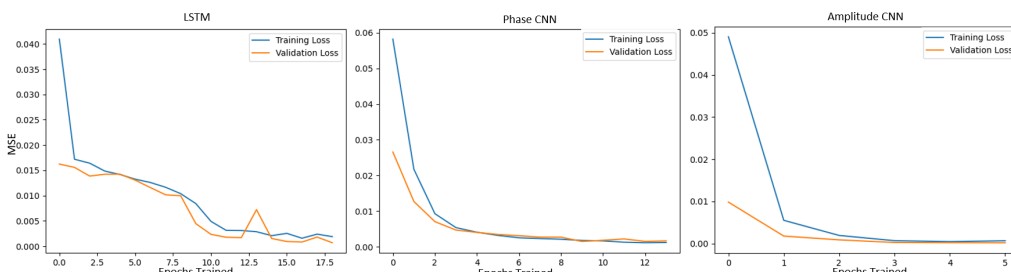

**Figure 15.** Comparison of training loss (mse) per epoch for LSTM, Phase CNN, and Amplitude CNN.

By examining the plot of real vs. predicted values and the regression line for the best fit in Figure 16, we can compare how accurately the models predict the acoustic path length. Note that at the same physical thickness, the acoustic path length differs depending on the driving frequency. We also examine the R-squared statistic to determine how well a regression line fits the predictions. For the LSTM model, we see there is less variance for the acoustic path lengths at 12.95, 64.75, and 123 radians meaning the model learned these more accurately than it learned the samples with acoustic path lengths of 25.13, 125.66, and 600 radians, and achieved an R-square score of 0.995. The variance across most acoustic path lengths for the phase model appears to be lower than that of the LSTM model; however, the variance is very high for the acoustic path length of 600 radians and the R-square score is only 0.971. The Amplitude CNN experiences the least amount of variance and thus has the best R-square score at 0.998.

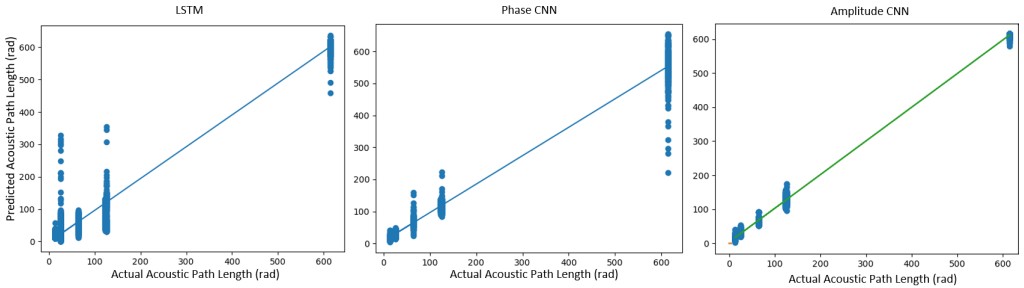

**Figure 16.** Actual acoustic path lengths (x) vs. predicted acoustic path lengths (y) with a regression line to demonstrate how close the model is to perfect accuracy. In each figure we see the predictions from left to right of: the path length 12.95 (1 MHz data), the path length 25.13 (1 MHz data), the path length 64.74 (5 MHz data), the path length 123 (1 MHz data), the path length 125.66 (5 MHz data), and then the path length 614.99 (5 MHz data). The derivation of these path length values are displayed in Table 4. It is immediately evident in the LSTM and Phase CNN graphs that the 1 MHz data are exhibiting less error in prediction than the 5 MHz data.

## 4. Discussion

The very low mean-square error of 0.00016 and very high R-square statistic of 0.998 exhibited by the Amplitude CNN were expected due to the high correlation between the amplitude values, low number of outliers, and similarities in the amplitude profile. The accuracy of the model can be explained by looking at the high levels of correlation both between the amplitude values at each lateral point and with the acoustic path lengths themselves in Figure 14. This high correlation coupled with the larger distribution of normal values seen in Figure 13 points toward data with few outliers and values which will be meaningful to the determination of acoustic path length. Additionally, examination of the transfer function amplitude profile of Figure 6 shows that the amplitude trends for samples exhibiting the same acoustic path length often exhibit similarities in the width of the peaks in the amplitude profile and strength of the signal despite having differently sized acoustic anomalies. These similarities are often difficult to discern immediately and require the review of many samples as the peaks may not always align due to very minor

changes in the placement of the specimen. All of these factors point to a model being highly likely to learn trends in the data despite the challenges of interpreting these data that are experienced by humans.

The increased mean-square error of 0.003 and reduced R-square statistic of 0.971 of the phase CNN model may also be explained by these figures. It is noted that there are fewer features that correspond to the acoustic path length and that the transfer function phase values are less correlated with one another. It is also clear from the box-and-whisker charts in Figure 13 that the phase data experience far less normal values and many outliers with a fairly large range. This seems to indicate that the phase data are behaving in a less predictable manner. This makes sense because the phase is a more complex function of x than the amplitude. Even in the Fraunhofer regime, the phase has ripples to either side of the central lobe. The phase profile in Figure 6 also supports this theory as the samples with corresponding acoustic path lengths rarely align in the phase profile.

The decreased performance of the phase model appears to be explained by the physics behind the model. The performance of the model can be discussed based on (a) diffraction and (b) the z-dependence of the radius of curvature. In point (b), the z-axis is along the propagation of the acoustic signal (perpendicular to the *x*-axis).

As to (a), the impact of diffraction on the lateral variation of the phase plays a role. We note that the x-dependence of the phase is much more complex in the Fresnel range than in the Fraunhofer range. Because of the complexity, a tiny error in the x-position upon the scanning procedure can lead to a huge difference in the phase value. The boundary between Fresnel and Fraunhofer ranges can be approximated by the aperture size, D, and the wavelength, lambda (see Figure 5). The data utilized by the models have two aperture sizes corresponding to the hole diameter of 2.17 mm and 6.47 mm. As the above vector Figure 5 indicates, the 5 MHz case makes the boundary 5 times greater than the 1 MHz case, meaning that the 5 MHz case diffraction is more Fresnel-like. At the same frequency, the 6.47 mm hole makes the boundary 9 times greater than the 2.17 mm hole. It is clearly seen in Figure 16 that the samples suffering the most are those with the acoustic path lengths of 64.736, 125.66, and 614.99. These are the samples that are collected at a frequency of 5 MHz. Because half of the data collected at this frequency are expected to fall in the Fresnel range, and the other in the Fraunhofer range, the data corresponding to these acoustic path lengths are likely to vary wildly in terms of the behavior of the wave. Because of the wildly different profiles, particularly of the phase data, which are more difficult to predict, one area of improvement may be to create an ensemble model for handling the phase data. This would mean training on CNN on Fresnel data and another on Fraunhofer data, then using the models together to make predictions on incoming data. This would allow more accurate predictions about the phase data while still relieving the operator from having to know whether the data were collected in a Fresnel or Fraunhofer regime.

As to (b), the z-dependence of the radius of curvature seems to affect the performance. The acoustic signal from the emitter is blocked by the hole due to the high reflectivity at the entrance of the hole. At the exit of the hole, a portion of the acoustic wave is missing over the diameter of the hole. According to Huygens' principle, we can view that the sensor signal is the superposition of the element waves on the exit plane of the disc. This means that the received acoustic signal is the superposed signal without a hole minus the actual superposed signal leaving the exit plane of the disc. Depending on the distance between the plane of the acoustic emitter and the end of the hole (i.e., the hole length), the radius of curvature of the missing wave at the disc's exit plane varies. It is likely that the greater the dependence of the curvature of the missing wave at the disc's exit plane on the hole thickness, the more sensitive the sensor signal to the hole thickness, i.e., it is easier to infer the hole thickness from the sensor signal. This can be examined by inspecting the Rayleigh length of the signals in relation to the thickness of the hole. The Rayleigh length is of importance as this gives the distance at which the wavefront reaches maximum curvature or minimum radius of curvature.

Figure 17 shows the above factors. Here, the left graph plots the physical hole thickness vs. hole thickness relative to the Fresnel-Fraunhofer boundary, and the middle and right graphs plot the physical hole thickness vs. hole thickness relative to the Rayleigh length for the hole diameter 2.17 mm and 6.47 mm cases, respectively. The left graph clearly indicates that the specimens with the larger anomaly are least Fraunhofer, meaning farthest from the Fraunhofer regime, while the small anomaly leads to data which are more Fraunhofer, meaning these specimens lead to diffraction which falls deep within the Fraunhofer regime. Particularly, the 1 MHz data with the small, 2.17 mm, anomaly are highly Fraunhofer while the 5 MHz data with the large, 6.47 mm, anomaly are far from Fraunhofer. The middle and right graphs indicate in all cases the hole thickness is orders of magnitude smaller than the Rayleigh length. Since the radius of curvature rapidly decreases with the z-distance toward the Rayleigh length, where the radius of curvature takes the minimum value, these two graphs depict the range where the radius of curvature (the phase) is sensitive to the z-distance (the hole thickness). The steeper the line, the more sensitive the phase is to the hole thickness.

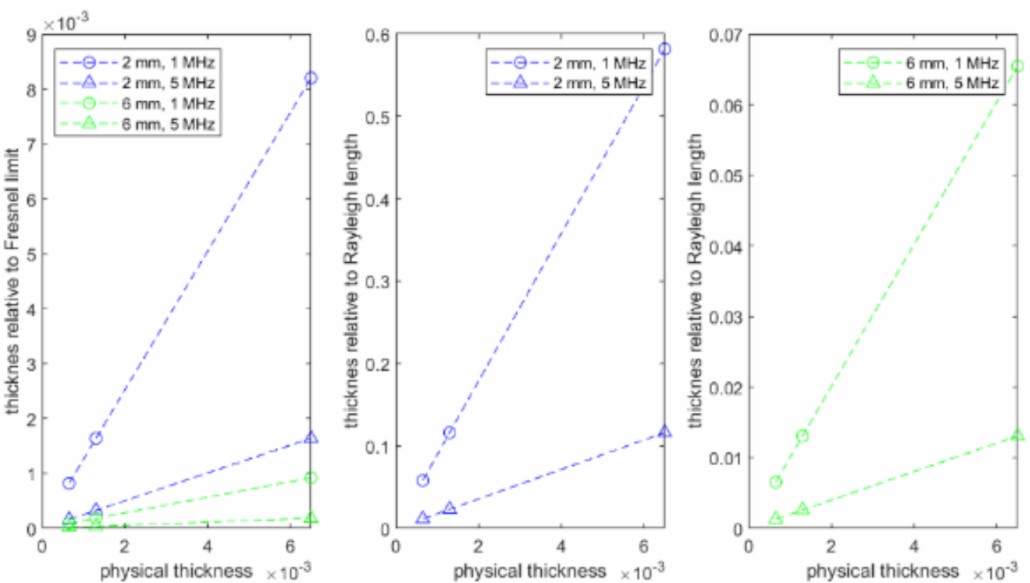

**Figure 17.** Hole-thickness vs. relative to Fresnel-Fraunhofer boundary (**left**), Hole-thickness vs. Rayleigh length for 2.17 mm hole diameter (**center**), Hole-thickness vs. Rayleigh length for 6.47 mm hole diameter (**right**).

The LSTM model performed between the phase and amplitude CNN models with a mean-square error of 0.00067 and an R-square statistic of 0.995. The advantage of this model is that it is able to use both phase and amplitude data and uses less preprocessing, as no images have to be formed. It is possible that this model is suffering the sample problems as the phase CNN suffers in that it may need to process Fresnel and Fraunhofer data separately. Similar to the future improvements for the phase CNN, this model may be split to create an ensemble in order to attempt to improve the performance to take advantage of the details held in both the phase and amplitude data. Another approach that may be taken on the LSTM model is to skip the preprocessing which calculates the phase and amplitude and allow the model to take in simply the transfer function of the wave. This would preserve some information that may be getting lost in the present model.

We test the theorized model improvements by using only the data taken with the 2.17 mm anomaly, as we know that this data predominantly fall in the Fraunhofer range, the only exception being the data with a path length of 614.99. In this case, the largest diameter is the thickness of the 6.47 mm disc, so the 2.17 mm data collected for this specimen are still in the Fresnel range. With only these data, the phase CNN and the LSTM model were again trained and tested to validate that removal of the Fresnel data leads to improved

ability to learn on the data. Using only these data leads to the phase CNN achieving a mean-square error of 0.0002 and an R-square statistic of 0.998; this is a large improvement from the previous R-square score of 0.971. Similarly, the LSTM model sees an improvement now achieving a mean-square error of 0.00027 and R-square 0.998. Figure 18 demonstrates the goodness of fit of the new predictions.

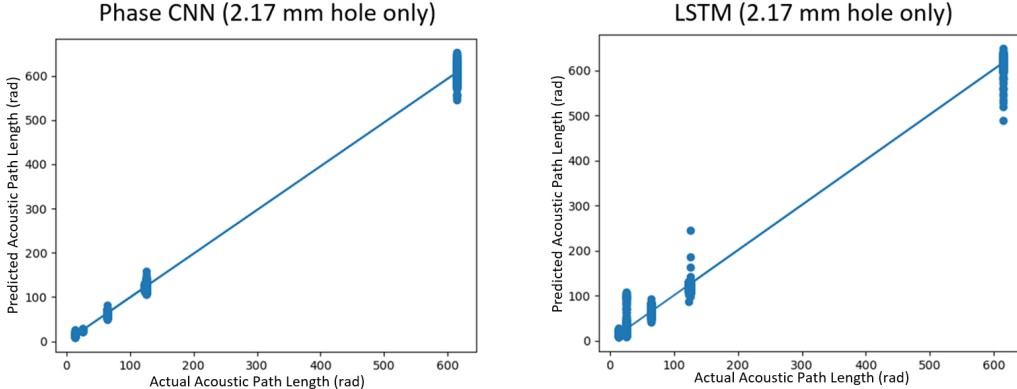

**Figure 18.** Actual acoustic path lengths (x) vs. predicted acoustic path lengths (y) on only the data with a 2.17 mm anomaly (Fraunhofer regime) with a regression line to demonstrate how close the model is to perfect accuracy plotted for both the phase CNN (**left**) and the LSTM model (**right**).In each figure, we see the predictions from left to right of the following: the path length 12.95 (1 MHz data), the path length 25.13 (1 MHz data), the path length 64.74 (5 MHz data), the path length 123 (1 MHz data), the path length 125.66 (5 MHz data), and then the path length 614.99 (5 MHz data). The derivation of these path length values are displayed in Table 4. Compared to the previous models, the error of both the 1 MHz and 5 MHz data is clearly reduced when only the more Fraunhofer data are used.

The LSTM model does not improve to the level of the CNN models, even with the simplified dataset. The LSTM is likely suffering from the additional complexity of leveraging both phase and amplitude data. As seen in the CNN models, the phase data fit best to a ReLU function, meaning it presents a linear trend, while the amplitude data fit better to a hyperbolic tangent, making the trend nonlinear. Further work is likely needed to determine how best to improve the LSTM model; however, Figure 19 depicts the large-scale improvement seen in the phase CNN when the Fresnel and Fraunhofer data are split. Here, it is evident that (1) the model when trained on only 1 MHz data is more accurate than when trained on just 5 MHz data if both the 2.17 mm and 6.47 mm holes are present and (2) when the 6.47 (more Fresnel-like data) is removed, both the 1 MHz and 5 MHz predictions are improved. (1) is consistent with the above argument that the 1 MHz data are more Fraunhofer-like than the 5 MHz data, and the sensitivity of the phase to the thickness is higher with 1 MHz (Figure 17). From (2), we can say that in the cases where both anomaly sizes are present, the data is compromised by the 6.47 mm data. This yields promising results in support of creating an ensemble of CNN models to separate the predictions of the more simplistic Fraunhofer and the more complex Fresnel data.

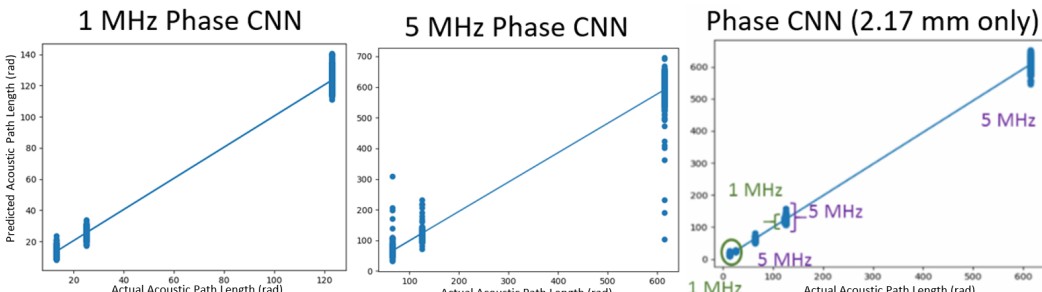

**Figure 19.** Comparison of performances of CNN Phase models for the original 1 MHz, 5 MHz datasets separately and the CNN phase model for 1 MHz and 5 MHz combined with 2.17 mm hole data only. In the left figure, we see the predictions of the model when run on solely the path length 12.95, the path length 25.13, and the path length 123, meaning only the 1 MHz data, which have the least range of diffraction patterns, as most of the data were Fraunhofer or Fraunhofer-like. In the middle figure, we see the predictions of the model when run solely on the path length 64.74, the path length 125.66, and then the path length 614.99, meaning only the 5 MHz data which have the most range of diffraction patterns, as much of the data were Fresnel or Fresnel-like. Finally, in the figure on the right we see the model performance when run on the 2.17 mm anomaly data from both the 1 and 5 MHz data, which clearly exhibits the least amount of error, as nearly all of the data generated with the 2.17 mm anomaly are within the Fraunhofer range, as seen in Table 3.

## 5. Conclusions

The performance of the CNN models and the LSTM model validate that both architectures are suitable for the approximation of acoustic path length based on changes in phase and amplitude data. Comparing the mean-square error of the models, the CNN for amplitude data is superior to the CNN phase and LSTM models. The next best model appears to be the LSTM model, which has the advantage of analyzing both phase and amplitude data. Additionally, examining the R-square measure for each model further solidifies this ranking of the models as the CNN for amplitude data achieved a near-perfect R-square statistic followed closely by the LSTM model and then by the CNN phase model.

Examining both the performance of the models and the physics behind the models, it is clear that there is room for improvement on the CNN phase model. This model performs poorly on the data collected at 5 MHz only. The data collected at this range encompass data at opposite ends of the Fresnel-Fraunhofer spectrum with some data being highly Fresnel and other data being highly Fraunhofer. As the expectations of phase and amplitude are different for the Fresnel range versus the Fraunhofer range, it may be difficult for the model to generalize data that falls into both ranges but corresponds to the same acoustic path length [18]. To overcome this, we propose future work on this model to extend it into an ensemble model where one model is trained in the Fresnel range and the other is trained in the Fraunhofer range. This should allow the model to pass input data to both models and accept the better outcome without forcing an operator to assess whether the data were Fresnel or Fraunhofer.

Another area for future improvements may be in running the LSTM on the raw waveforms. The present LSTM interprets only the transfer function phase and amplitude for the driving frequency. While this should be enough to make decisions about the data, it may not be enough to paint a clear picture of the system. We may experience a loss of information when the data are cut down to a single frequency range, so keeping and analyzing the full spectrum may provide better approximations of acoustic path length. For the LSTM, it is also worth considering if the phase and amplitude data should be analyzed separately as the CNN models found that phase was better predicted with a regression to the ReLU function while amplitude was better predicted by fitting to the hyperbolic tangent function. Further variations of the LSTM model should be created and evaluated to determine if increasing the data leveraged and splitting the data sources leads to better accuracy.

**Author Contributions:** Conceptualization, S.Y.; methodology, B.E.J. and S.Y.; software, B.E.J.; validation, B.E.J. and S.Y.; formal analysis, B.E.J. and S.Y.; investigation, B.E.J. and S.Y.; data curation, B.E.J.; writing—original draft preparation, B.E.J.; writing—review and editing, B.E.J. and S.Y. All authors have read and agreed to the published version of the manuscript.

**Funding:** This research received no external funding.

**Institutional Review Board Statement:** Not Applicable.

**Informed Consent Statement:** Not Applicable.

**Data Availability Statement:** Data collected in the experiments and used in this paper can be downloaded at: https://drive.google.com/drive/folders/1zCaapPp3Rwu-Uc-8Co-DluGEuFmtAOjh?usp=sharing (accessed on 4 February 2023).

**Conflicts of Interest:** The authors declare no conflict of interest.

## Abbreviations

The following abbreviations are used in this manuscript:

| | |
|---|---|
| AE | Acoustic Emissions |
| BVID | Barely Visibile Impact Damages |
| CNN | Convolutional Neural Network |
| CSV | Comma-Separated Value |
| EPS | Extracellular Polymeric Substance |
| FFT | Fast Fourier Transform |
| LSTM | Long Short-Term Memory |
| SDFT | Short Distance Fourier Transform |

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
