# Peer review of "Deep Machine Learning for Path Length Characterization Using Acoustic Diffraction"

_applsci, doi:10.3390/app13052782_

Round 1

Reviewer 1 Report

In the present study is proposed the use of deep machine learning for acoustic path length characterization in metallic specimens by means of Long Short-Term Memory and Convolutional Neural Network learning algorithms.

Although the subject is interesting, some parts of the text require multiple reads to try to understand their meaning, namely in the introduction section. Description of theoretical concepts and details about experimental tests must be significantly improved before publication.

General comment:

As mentioned in the manuscript by the authors, the motivation of the paper deals to the necessity to produce a methodology for assessing the health of biofilms in situ. The work developed, namely the use of metallic rings as experimental samples and consequently the analyse of the collected signals from those samples is quite different than a problem related to biofilms with very little thickness.

The authors must clarify this main point of the work.

Detailed comments about the manuscript:

1. Page 4, line 6

“…100µM…”, should be replaced by“…100 µm …”.

2. Page 4, bottom

The authors say: “The acoustic impedance here is determined by the properties of air, steel, and water at room temperature since these are the media through which the acoustic wave passes”.

The materials used in the experiments should be mentioned previously.

Why the properties of water? Was used in the presented work?

3. Page 5, table 1

The same previous comment

4. Page 5, line 5

Why use the reflection coefficients for energy instead of pressure? Please clarify.

5. Page 5.

In equation (3) should be approximately, instead of equal.

6. Page 5

The authors say: “From the above calculations, we are able to infer that upwards of 88% of the wave will be reflected at the anomaly”

Why? The anomaly is water? If the anomaly is like the one in Fig. 1 the equation (1) cannot be applied.

7. Page 5

The authors say: “This means that much of the wave will be bending around the obstacle at the center, which will lead to evidence of an acoustic shadow behind the hole”.

This sentence must be clarified, a figure with details of setup (transducers, media...) should be provided, otherwise this sentence is not understandable.

By other side the authors speak about “hole”. If it is a hole, should be air inside? Again, why was considered water in previous sentence?

8. Page 5

The authors say: “Data in this experiment is captured with two wavelengths and 2 possible anomaly sizes and always observed at a distance of 6.47 mm”.

The size of anomalies should be mentioned.

9. Page 5

The authors say: “As we see in the figures, theoretically, the acoustic shadow should be apparent in both of the 1 MHz cases but may be more difficult to observe or even missed in the 5 MHz cases”

In not clear in the figure where the mentioned shadow.

10. Page 5, Fig. 2

A lot of doubts arise from this figure. As mentioned in the previous comment, is not clear the content of the fig. 2. Black box are the anomalies? What are the vertical traces and the triangular gray forms?

There are 2 different frequencies and to 2 wavelengths that give rise to two different propagation velocities (velocity=wavelength*frequency), why is that? The figures evolves to different materials?

What is the relation of fig. 1 and fig. 2?

Additional information should be provided to understand this figure

11. Page 6, line 1

In the fig. 3 the authors mention hole. In the text this should also be mentioned. How were the figures obtained? Formulations should be provided. Or were taken from literature? If so, reference must be provided.

12. Page 6, Fig. 2

The radiation patterns are equal for the 4 different situations presented? I think that this no does not make sense. Please revise the figure.

In the legend the authors say that the relationship of the observation point to R are displayed. Where?

13. Page 7, Fig. 4

The authors speak here in transfer function. Why this transfer function was not defined previously?

Concerning the figure 4 itself:

1 - Both YY axes have no units.

2 - The legend are not understandable (PL ?. D?).

3 - More details about how the figures here obtained must be provided (phase and module of transfer function).

14. Page 7, line 1

The authors say: “As discussed in [9], additional complexity is introduced as the signal leaves the transmitter and enters into the system”.

Improve the sentence, special when saying “…enters into the system…”.

15. Page 7, line 4

The authors say: “…wave’s transfer function…”.

Is the same as transfer function?

16. Page 7

When the authors speak of the materials and methods, information about the probes used (supplier, models) must be provided. Also, information about how the coupling of probes and samples is done, must be provide.

17. Page 7

The authors say: “The acoustic wave used to probe the specimen is emitted from directly below with the specimen’s hole over the center of the transmitter”.

What is the transmitter probe dimension?

18. Page 7

The authors say: “…the samples are collected …”.

The authors want to say "…the signals are collected…"?

19. Page 8, Fig. 5

The figure 5 must be improved, the differences of the samples is not clear.

How many samples are in total?

20. Page 8

The authors say: “The 1 MHz transducers have a beam waist of 12 mm and the 5 MHz transducers have a beam waist of 7 mm”.

Waist? "Beam width" is more adequate?

21. Page 8, Fig. 6

The figure 6 must be improved, in not clear da position of the probes in relation to the sample.

22. Page 8

The authors say: “The oscilloscope software stores each waveform…”

The oscilloscope stores? Or the computer connected do the oscilloscope stores?

23. Page 9

The authors say: “…5 samples are captured…”.

The authors mean to say “…5 signals are captured…”?

24. Page 9

The authors say: “…of 60,000 sets of waveforms per acoustic path length…”

More detail, for instance in a table, should be provided to clarify how this value were obtained, the text is messy.

25. Page 9

The authors say: “This allows us to remove a degree of noise from the phase and amplitude features in our training please clarify how this process remove the noise”.

Please clarify how this process of division of the transfer functions can remove the noise.

26. Page 16 to 20

Figures from 13 to 18 should be improved (fonts to small) and units in the axis should be provided.

27. Page 17

The authors say: “…examination of the transfer function amplitude profile of Figure 4 shows that the amplitude trends for samples exhibiting the same acoustic path length are fairly similar…”

Looking to fig 4 in not possible to see a similar trend, some details about explaining this sentence are needed. It was a mistake?

28. Page 19, line 3

The authors speak here about “Rayleigh length”. Until now the authors do not speak about Rayleigh length, a note about this concept must be introduced.

How Rayleigh length were calculated for the different situations?

29. Page 19

The authors say: “The left graph clearly indicates that the specimens with the larger anomaly are least Fraunhofer while the small anomaly leads to data which is more Fraunhofer”.

Clarify this sentence, especially the terms: "least Fraunhofer" an "more Fraunhofer".

Author Response

We have addressed your concerns. Please see the attachment for details.

Reviewer 2 Report

The paper is very good, and I only have minor changes/suggestions. See the attached file.

Author Response

Thank you for your review. We have addressed your concerns in the following ways:

1) The definition of the axes has been added for clarification.

2) The small text in Figure 3 has been addressed and the description of this figure has been expanded.

3) In Figure 15, a description of the groupings has been added along with a table for reference.

4) In Figure 17, a description of the groupings has been added which also references the new table.

In Figure 18, a description of the groupings has been added with references to the diffraction figure for a better explanation of the trend shown.

Round 2

Reviewer 1 Report

The article is now ready to be published.